# Wearable Sensors for Activity Recognition in Ultimate Frisbee Using Convolutional Neural Networks and Transfer Learning

**DOI:** 10.3390/s22072560

**Published:** 2022-03-27

**Authors:** Johannes Link, Timur Perst, Maike Stoeve, Bjoern M. Eskofier

**Affiliations:** Machine Learning and Data Analytics Lab, Department Artificial Intelligence in Biomedical Engineering, Friedrich-Alexander-Universität Erlangen-Nürnberg (FAU), 91052 Erlangen, Germany; timur.perst@fau.de (T.P.); maike.stoeve@fau.de (M.S.); bjoern.eskofier@fau.de (B.M.E.)

**Keywords:** inertial measurement unit, activity recognition, sensor-signal-based machine learning, convolutional neural network, deep learning, wearable sensors, marginal sports, transfer learning

## Abstract

In human activity recognition (HAR), activities are automatically recognized and classified from a continuous stream of input sensor data. Although the scientific community has developed multiple approaches for various sports in recent years, marginal sports are rarely considered. These approaches cannot directly be applied to marginal sports, where available data are sparse and costly to acquire. Thus, we recorded and annotated inertial measurement unit (IMU) data containing different types of Ultimate Frisbee throws to investigate whether Convolutional Neural Networks (CNNs) and transfer learning can solve this. The relevant actions were automatically detected and were classified using a CNN. The proposed pipeline reaches an accuracy of 66.6%, distinguishing between nine different fine-grained classes. For the classification of the three basic throwing techniques, we achieve an accuracy of 89.9%. Furthermore, the results were compared to a transfer learning-based approach using a beach volleyball dataset as the source. Even if transfer learning could not improve the classification accuracy, the training time was significantly reduced. Finally, the effect of transfer learning on a reduced dataset, i.e., without data augmentations, is analyzed. While having the same number of training subjects, using the pre-trained weights improves the generalization capabilities of the network, i.e., increasing the accuracy and F1 score. This shows that transfer learning can be beneficial, especially when dealing with small datasets, as in marginal sports, and therefore, can improve the tracking of marginal sports.

## 1. Introduction

Monitoring athletes has been in the interest of players and coaches for a very long time. As early as the 1930s, two Germans used heart rate observations of middle-distance runners to improve the training procedure of their athletes. The proposed interval training has been adopted worldwide and has enabled athletes to set new records. The hunt for new training strategies based on scientific data of athletes has continued ever since [1]. Traditionally, monitoring was done by manually observing the athletes, which was time-consuming and needed many experts in the field. With the increasing computational capacities of computers, smaller and more affordable hardware, and the development of powerful algorithms, monitoring athletes has become feasible for a broad spectrum of protagonists [1]. A highly objective and reliable qualitative analysis using wearable sensors is an accepted alternative to traditional lab-based assessment. Wearable sensors are portable, low-cost, easy to use, and usually do not limit athletes in their movement [2]. Their success has opened the path for commercial use, for example, at pro soccer clubs, whose players wear sensors even during competitive matches [3]. However, the development of more advanced systems classifying distinctive movement patterns is still an active research topic [2].

Another important goal of monitoring athletes is the prevention of injuries. Even though most acute injuries occur in contact sports like American football, injuries resulting from overuse often happen in non-contact sports. A study amongst collegiate athletes revealed that over 29% of the injuries result from overuse, with a much higher estimated number of unreported cases [4]. Fine-grained monitoring of players’ actions can help to detect high training loads for specific body parts, which have been linked to injuries, thus allowing the coach to adjust the training individually [5].

In the literature, a great variety of sport-specific activity recognition systems exists, which can be used for fine-grained, athlete-specific monitoring [6]. To detect actions reliably, a sufficient amount of training data needs to be recorded, which is not only cumbersome but sometimes not possible at all, especially in marginal sports. To combat the lack of data, transfer learning approaches use data from another domain for pre-training. In a second step, model parameters are fine-tuned on the target dataset [7].

Our research contributes in the following ways. Firstly, we developed the first activity recognition system for Ultimate Frisbee. Therefore we trained a Convolutional Neural Network CNN to distinguish seven different throwing techniques plus catches. Secondly, we pre-trained the same architecture on an existing volleyball dataset [8] to investigate the potential of transfer learning for marginal sports. Thirdly, we explored the generalization capabilities of transfer learning when dealing with small-scale datasets.

A graphical summary of the procedure in this work is shown in Figure 1.

## 2. Related Work

### 2.1. Sensor Based Human Activity Recognition in Sports

The scientific and commercial interest in HAR has been growing in recent years, leading to extensive research [9]. Traditionally, vision-based solutions have been used for activity recognition. However, those come with the disadvantages of being expensive, causing privacy issues, and having to be mounted at specific positions, which may not be feasible for minor sports outside of huge stadiums [10]. On the other hand, sensor-based systems using IMUs like accelerometers, gyroscopes, and magnetometers have been adopted in the research community [6]. Historically, accurate sensors had to be custom-built for each application. Smaller, cheaper, and better availability of sensors have made it possible to analyze numerous different sports and movements in recent years [9]. Cust et al. [6] provide a broad overview of machine and deep learning for sport-specific movement recognition. Their review includes a comparison of several sensor- and vision-based solutions for HAR in sports. Even though the respective authors used many different methods for their HAR systems, the general workflow included: preprocessing, segmentation, feature extraction, dimensionality reduction, and classification. During the preprocessing step, many authors propose low pass filters to remove unwanted noise [10]. There has been successful work with transforming the signal into the frequency domain, such as applying a wavelet transformation [11] or a fast Fourier transform [12]. The trend for the classification task is more and more towards neural networks, specifically to CNNs and long short-term memory (LSTM) [13], due to their good performance. CNNs have been initially used for image detection tasks because they preserve the spatial information of neighboring pixels [14]. Time series data also include spatial information between subsequent measurements [15]. In addition, deep learning-based approaches have the advantage of skipping the feature extraction step. Chen et al. [16] proved that CNNs can compete with other algorithms, reaching an accuracy of 93.8% when classifying eight tasks of daily living.

By now, the research community has developed a variety of classification pipelines that specifically target one discipline of sports. Anand et al. make use of the better availability of sensors and use smartwatches to classify strokes in swing sports, like tennis or badminton, and give the players feedback on their technique [17]. Using a bi-directional long short-term memory network, they achieve accuracies between 78.9% and 94.6% depending on the swing sport. Brock et al. classify common errors of ski jumpers and chose IMUs over a vision-based solution to account for bad weather conditions and to get numerically comparable data [18]. They achieve error recognition rates between 60% and 75%. Kautz et al. classified actions from beach volleyball players using data from a wrist-worn accelerometer. Amongst others, they programmed a deep convolutional neural network, which reached an accuracy of 83% [8].

### 2.2. Transfer Learning

Transfer learning refers to the technique of using knowledge gained in one domain (source domain) and applying it in another domain (target domain). Its advantages over training a classifier without prior knowledge include shorter training durations and the ability to solve previously unsolvable tasks.

Transfer learning has been applied to CNNs in various domains. This includes, among others, image classification [19,20,21,22,23] and image segmentation [24,25]. In order to improve their accuracy, some of the studies used several data pre-processing and augmentation techniques.

Apart from this, the review by Cook et al. [26] provides a broad general overview of transfer learning for activity recognition. Instance-based transfer learning can be used when the source and target domain are the same or very similar, but the task differs. Training samples are reweighted and directly fed into the classifier for the target task [27]. Tommasio et al. [28] use a form of parameter transfer where they split the parameters of a Support Vector Machine (SVM) *w* into two parts, where one is shared by the two tasks. Fawaz et al. [29] investigated transfer learning for time series classification and came to the conclusion that the choice of the source dataset heavily influences the success or failure of a transfer learning-based approach. Better generalization capabilities can be reached with a suitable dataset for the target task. There has also been work on transfer learning applied to HAR. Morales et al. [30] were the first with the idea of transferring low-level features, learned in the lower layers of a CNN for an HAR task.

## 3. Materials and Methods

The following chapter describes the setup and methods used in this work. First, the activity detection pipeline is explained, which is used to extract relevant samples from the continuous sensor data stream. Then, the labeling procedure and the utilized augmentation techniques are presented. Next, the classification pipeline is described. The classifier was trained from scratch and compared to a network pre-trained on a beach volleyball dataset as a source domain. This transfer learning approach is explained, followed by the evaluation techniques.

### 3.1. Datasets

This section describes the two datasets used as a source and target dataset. Furthermore, we will explain how the studies to record the data were conducted and how actions were detected. The description of the volleyball dataset by Kautz et al. [8] is followed by a short introduction to Ultimate Frisbee. Afterward, we present the frisbee dataset recorded for this work.

#### 3.1.1. Volleyball Dataset

This dataset consists of three-dimensional acceleration data of common actions in beach volleyball. It was acquired by Kautz et al. for their paper on “Activity recognition in beach volleyball using a Deep Convolutional Neural Network” [8]. Each of the 30 participants was outfitted with a wrist-worn accelerometer, which sampled at 39 Hz. The signal was recorded with 14 bits per axis and truncated at ±16 g. Note that the axis x, y, and z refer to the sensor’s coordinate system. There was no transfer to real-world coordinates. The players’ experience ranged from beginners to professional level athletes. The overall goal was to develop a recognition and classification system which extracts relevant sections from the continuous input and assigns them to a class. The classification incorporates ten different volleyball activities, including, among others, the underhand serve, block, and dig. Table 1 shows a summary of the two datasets.

The first step of the processing pipeline, the detection of potentially relevant signal segments, was done directly on the worn microcontroller. Since relevant actions include a ball contact, the idea was to detect the high-frequency spikes in the acceleration data. This was achieved by passing the signal through a high-pass filter, computing the $L_{1}$ norm, and smoothing the signal using a low-pass filter. If the signal does not exceed a certain threshold, it is rejected. However, this approach was not sufficient to detect actions accurately. To further refine the process, the authors calculated the swing movement by averaging the absolute acceleration in all three directions for 200 ms before the impact. With the swing movement and the amplitude at the peak, Kautz et al. trained a decision tree to discard irrelevant actions. For a more detailed description of the action detection pipeline and the recorded dataset, see the original publication [8]. We use the extracted actions of interest from the dataset for our work.

The classification task used multiple feature-based classifiers: support vector machine, k-nearest-neighbor, Gaussian naive Bayes, decision tree, random forest, and VOTE as a meta classifier. A deep convolutional neural network (DCNN) outperformed all classifiers with hand-crafted features.

#### 3.1.2. Ultimate Frisbee Dataset

In recent years, Ultimate Frisbee has been one of the fastest-growing sports globally. It has its origins in the 1950s in the United States and has since spread worldwide [31].

Ultimate Frisbee is played seven vs. seven on a field with the length of a soccer field and half its width. There are 15 to 18 m deep endzones comparable to American Football at both ends. A player can score a point for their team by catching the frisbee in the endzone. Teammates can pass the frisbee to each other as they please. After a catch, there is a ten-second time limit to get rid of the frisbee. A failure to do so, or an incomplete pass, results in a turnover. Travelling is—except for a basketball-style pivot step—forbidden. As a non-contact sport, every kind of physicality is illegal.

This work focuses on three basic throws: backhand throws, forehand throws, and overheads—sometimes called a hammer. Figure 2 shows a sketch of the basic throwing techniques. Depending on the release angle, the frisbee moves on a different trajectory. Therefore, forehand and backhand throws are further separated by the frisbee’s path: flat, outside-in, and inside-out. An example of the variety of backhand throws by a right-handed player is depicted in Figure 3. Table 2 shows a description of all throwing techniques together with the number of samples per class contained in our frisbee dataset.

Since we wanted our classifier to distinguish between flat, outside-in, and inside-out throws, which require players to have an advanced level of skill, we recruited most of the players from a local Ultimate Frisbee team. We recorded 14 participants (12 male, 2 female) who throw mainly with their right hand. The mean age of the players was 31 years (standard deviation: 7 years). Everyone except one participant reports at least four years of experience in Ultimate Frisbee (mean: 11 years). Except for one player, everyone has played Ultimate Frisbee competitively in any form of club or association. Two players have participated in the world championships.

Each player was outfitted with an IMU (Portabiles NilsPod) at the wrist of their dominant hand (see Figure 3). The sensor initially sampled at 512 Hz and recorded acceleration and gyroscope data in all three dimensions. The sensors write the recorded data to the built-in memory. The measurement range for the accelerometer is ±16 g and $\pm 2000^{\circ }$ per second for the gyroscope.

One recording session featured two players at once. After a couple of throws to warm up, the players positioned themselves at a predefined distance. The distances visually marked by pylons were ten meters in round one, 15 m in round two, and 25 m in round three. Some of the participants also did a fourth-round with 30 m distance. Each player threw each kind of throw at least five times in each round. Table 2 lists the types of actions and the number of occurrences in the dataset. For labeling the activities, the data acquisition was filmed using a tripod-mounted wide-angle action camera.

### 3.2. Activity Detection

In order to classify the different throws, potentially relevant actions had to be detected first. Comparing the acceleration and gyroscope plots with the reference video reveals that actions involve high-frequency peaks in the plots. Furthermore, it was essential to exclude another ordinary and somewhat similar-looking but uninteresting activity: Running to retrieve a not caught frisbee. For the action detection, the gyroscope data was disregarded. For this purpose, we first calculated the acceleration norm and used the z-score algorithm to detect peaks [32]. We used a window size of 1.6 s, a threshold of 3.5, and an influence of 0.5.

During continuous actions (like running), the moving standard deviation of the signal is higher than when the subject is in an idle state. This property helps to more accurately detect throws because they usually follow a situation where the player is, by rule, not moving. It can also detect catches because even though the participant might run to make a catch, the frisbee’s abrupt deceleration transfers much energy to the player’s hand and thus creates very high-frequency peaks.

Since this first step has the issue of recognizing multiple peaks for only one action, peaks were forced to be at least 400 samples (0.78 s) apart. If this condition was not met, only the first peak gets included due to the assumption that the first high-frequency peak indicates the initial action, whereas the latter ones are only post-pulse oscillations. The gyroscope data were disregarded for the activity detection and only used during the classification. After the peak detection, we resampled the IMU data to 39 Hz to match the volleyball data.

### 3.3. Augmentation Techniques

In order to get a more robust classifier and achieve better generalization results, we employed three random-based augmentation techniques. Each peak corresponds to one detected action, and the classifier works with samples of a fixed length. The continuous stream of acceleration and gyroscope data is cut into windows of a fixed length which serve as an input feature vector $x\in R^{6\times l}$ with *l* being the number of samples in the window. The window starts 0.8 s before the peak and lasts until 1.0 seconds after the peak (l=(0.8s+1.0s)·39Hz). Augmented samples are then generated by adding three randomly moved windows per detected action. The maximum shift of the peak thereby is 0.2 s.

Furthermore, we tripled the number of samples by rotating existing ones around a random angle in 3D space. During the training phase, a normally distributed noise is added to each channel as the third augmentation technique. All these techniques are applied to both the source and the target dataset.

Additionally, we rebalanced the datasets by a combination of the synthetic minority over-sampling technique (SMOTE) and a clean-up by the edited nearest neighbor (ENN) algorithm. Since SMOTE can use marginal outliers, which generates unsatisfactory training samples, ENN is used to remove the generated samples, which are in the feature space far away from the majority of the class [33,34].

### 3.4. Classification

For classification, a CNN was developed. The architecture was partly adapted from Fawaz et al. [29]. It consists of two input layers: One for the acceleration data and one for the gyroscope data. Three convolutional layers follow each input layer. Batch normalization is used after each convolutional layer in order to reduce the training times and achieve better results [35].

The widths of the kernels of the convolutional layers are from the first to the last layer: eight, five, and three. After the convolutional layers, a global maximum pooling layer is used. Global average pooling has the same effect on the accuracy but has severe disadvantages concerning the training times [36]. In contrast to the findings of Boureau et al. [36], no improvement in accuracy could be observed when using maximum pooling. The results of the pooling layers are concatenated and passed via one fully connected layer with 64 neurons to the output layer. The convolutional and fully connected layers use the rectified linear unit (ReLU) as an activation function. The last layer uses the softmax function to assign probabilities to each class. The architecture of the network is depicted in Figure 4. Additionally, in Table 3 the network architecture is described in detail.

During the training procedure, an optimizer implementing Adam’s algorithm is used [37]. Even though Adam computes individual adaptive learning rates for different weights, the implementation has shown more stable results when gradually decreasing the initial learning rate η over the epochs. Therefore, we used early stopping with a patience of 10, restoring the best weights and reducing the learning rate via the inverse decay function. The training was performed on an NVIDIA GeForce RTX 2080 Ti.

We use a leave-one-subject-out cross-validation for network performance evaluation investigating the accuracy, and the macro averaged F1 score, which combines the class-wise F1 scores. Therefore, we investigate the accuracy, F1 score, and training time per fold averaged over 14 folds.

### 3.5. Transfer Learning

The volleyball dataset obtained by Kautz et al. [8] served as the source domain in our transfer learning approach. Participants of both studies were equipped with a comparable sensor at the same position at their wrist. Kautz et al. [8] have already done the activity detection for the volleyball dataset, which means that our work is based on annotated samples of a fixed length that contain relevant actions from the domain. At first, the network (compare Figure 4) is trained on the volleyball dataset. Since this dataset lacks gyroscope data, only one input layer and subsequent convolutional, normalization, and pooling layers are used. During each run, the samples of 25 participants serve as training data, and the samples of two participants as validation data. The samples of three participants are used as test data. The weights of the convolutional layers from the best fold are saved and will be reused for subsequent transfer learning approaches.

The model is evaluated using a 3-fold-cross validation following the work of the original authors, which ensures good comparability. Note that Kautz et al. [8] used a slightly different architecture than what is used in this work.

The second step of the transfer learning approach is to train the actual network on the frisbee dataset. Therefore, the previously saved weights of the convolutional layers are loaded into the network. Depending on the exact configuration, those layers may be frozen, which means their weights will not be updated during the backpropagation phase. Due to the sharp peaks in gyroscope and acceleration data, we assume that the kernels for both network inputs are similar in the first convolutional layers. Therefore, we use the pre-trained weights for both acceleration and gyroscope data even though only the acceleration data was used for the training.

The remaining layers are initialized using Glorot’s initializer to combat the problem of vanishing gradients. Glorot et al. [38] showed that when going through the layers of a neural network, the variance in the layer’s output increases, and thus the gradients for the lower layers become very small when initializing the weights with a Gaussian normal distribution. Thus, the lower layers are only receiving minimal to almost no updates [39].

### 3.6. Experiments

The classification task on the frisbee dataset was conducted using both the detailed and aggregated labels. The first task features nine distinct classes (see Table 2). The latter one features five classes aggregating the different forehand and backhand throws. We then look at the performance of transfer learning, varying the number of pre-trained and frozen layers. Finally, we will resolve whether transfer learning can improve generalization capabilities when fewer training samples are available. This represents a smaller dataset due to a smaller study design. It might be cumbersome to acquire study participants in marginal sports, especially for individual sports. For comparability to the previous experiments, we did not decrease the total number of participants used for training but instead omitted the data augmentation steps. In general, a decrease in performance is expected for smaller datasets.

## 4. Results

The following paragraphs present the results obtained from the experiments described in the previous chapter. We start with the results of our architecture on the volleyball dataset, since its performance influences the performance of the transfer learning experiments. We then present the results of the frisbee dataset and the transfer learning results on the network’s performance.

### 4.1. Volleyball Dataset

The leave-three-out cross-validation on the volleyball dataset resulted in a mean accuracy of 86.6% over all folds surpassing the accuracy of Kautz et al. [8] by 3.45 percentage points. The average F1 score is 68.7%.

### 4.2. Frisbee Dataset

The training from scratch on the frisbee dataset yields an overall accuracy of 66.6% and a macro averaged F1 score of 52.3%. Figure 5 shows the confusion matrix of the classification performance. The values are normalized along the true classes. The confusion matrix clearly shows block matrices for the different forehand (classes 0 to 2) and backhand (classes 3 to 5) throws.

Furthermore, some overheads (class 6) have been misclassified as one of the forehand throws (3 to 5).

If we drop the distinction between flat, outside-in, and inside-out throws and focus on the aggregated classes backhand and forehand, the leave-one-subject-out cross-validation results in a mean accuracy of 89.9% and a macro averaged F1 score of 88.4%. The corresponding confusion matrix is shown in Figure 6.

### 4.3. Transfer Learning

As described previously, transfer learning is realized by freezing or fine-tuning the weights of the first few layers after the initial training of the network on a source domain.

Figure 7 shows the mean training time per fold, F1 score, and accuracy for different training configurations. The training time is measured, including balancing operations and the patience of early stopping. Using the pre-trained weights for all three convolutional layers, but without freezing these layers, the accuracy drops from 66.6% with randomly initialized weights to 66.2%. In contrast, the F1 score slightly improves to 52.9%. The training time per fold decreases from 217.6 s to 198.6 s.

An increased number of frozen layers leads to a decrease in both accuracy and F1 score. While the training time per fold is only slightly shorter for one layer frozen (205.2 s) for the two layers initialized and frozen, the training time is considerably shorter (119.7 s).

### 4.4. Transfer Learning with Smaller Dataset

Looking at the transfer learning performance using the reduced training dataset, we generally see a drop in performance compared to the network trained on the entire dataset, i.e., with all data augmentations. Figure 8 shows the mean training time per fold, F1 score, and accuracy for this training scenario. With randomly initialized weights, the network achieves an accuracy of 52.0% and an F1 score of 31.8%, which takes 8.6 s per fold. Using the pre-trained weights for all three convolutional layers and not freezing them, the accuracy increases to 60.5% and an F1 score of 36.8% while also increasing the training time to 9.8 s. Having one or two layers initialized and frozen increases accuracy and F1 score compared to the random initialization. The performance with random initialization of weights is slightly worse than without freezing the layers. However, it leads to shorter training times.

## 5. Discussion

The goal of this work was to create the first HAR system for different throws in Ultimate Frisbee using a convolutional neural network and study the effects of transfer learning from a beach volleyball action recognition task.

### 5.1. Study

The data recording took place at the end of March 2021, where the participants who practiced regularly multiple times a week were not allowed to do so for roughly five months due to COVID-19 restrictions in Germany. Even though most of the players met privately to practice throws, the intensity and complexity of the throws increases during competitive matches. The players were aware of this issue since they reported that the lack of practice negatively influenced their throwing capabilities. This decline in capabilities is not only a perceived effect as Korkmaz et al. [40] have shown. Their study conducted with amateur football players found that the months-long detraining process during the pandemic led to a significant deterioration in players’ physical and motoric abilities.

The data have been recorded with only two players at a time in a controlled environment. Future work has to investigate whether the proposed pipeline works under competitive conditions during a match. Stoeve et al. [41] investigated the transferability of a football activity recognition pipeline from controlled conditions to real-world scenarios. They observed a decrease in performance for a feature-based approach, whereas performance was comparable for the proposed CNN, thus indicating good generalization to complex scenarios [41].

### 5.2. Activity Recognition

The network achieved an accuracy of 89.9% in the 5 class problem without favoring the majority class, which supports the evidence that CNNs work well for time series classification. To put these numbers into perspective, Anand et al.’s CNN for stroke classification reached accuracies between 77.2% (badminton) and 93.8% (tennis) [17].

The decreased performance for the 9 class problem shows that separating flat, outside-in and inside-out throws is a tough challenge. The confusion matrix (Figure 5) reveals block matrices for the different forehand and backhand classes. Inside these sub-matrices, the error rate of the classifier was very high, the network did not predict reliable results, and one could argue that the classifier can not distinguish the detailed classes. Due to better applicability in a real-world scenario using smartwatches or fitness trackers, we fixed the IMUs at the wrist of the dominant hand. However, future work should consider placing the sensor at the back of the hand since the subtle differences in movement between performing flat, outside-in, and inside-out throws originate mainly from the wrist’s movement. A sensor placed above the wrist has disadvantages in recording these small changes and, therefore, worsens the classifier’s performance. Since many frisbee players wear gloves for better friction, a sensor could be fixed on the glove.

We also tested the classification using other architectures, including a ResNet architecture [42]. The tested networks achieved lower or, at most, comparable accuracies. The disadvantage of the architectures achieving the same accuracy as, e.g., the ResNet architecture is the much longer training time. In our case, using only three ResNet blocks already doubles the training time.

### 5.3. Transfer Learning

With the transfer learning approach, the accuracy and F1 score could not be improved. The reason for this is probably, as already mentioned, that the minor differences between the detailed forehand and backhand throws are not represented in the IMU data.

Nevertheless, transfer learning decreases the training time while maintaining similar performance results. Therefore, depending on the individual application and focus, it might be beneficial to use transfer learning to achieve faster training times.

One drawback of the employed source dataset is the missing gyroscope data. We used the weights of the convolutional layers pre-trained on the acceleration of the volleyball data as initialization for the layers of the frisbee gyroscope data. Since we achieve comparable results, our assumption that both acceleration and gyroscope have similar low-level features stands. While the focus on only acceleration data may have been sufficient for classifying beach volleyball actions, investigations including multiple architectures revealed the necessity of including gyroscope data for frisbee activity recognition. Therefore, disregarding the gyroscope data has never been an option in our research.

### 5.4. Transfer Learning with Smaller Dataset

As expected, reducing the dataset leads to a general drop in performance in all training configurations. However, transfer learning improves both accuracy and F1 score in all tested training configurations. The training time slightly increases or decreases depending on the exact transfer learning setup. Overall, the results show that transfer learning can improve the network’s generalization capabilities when using small datasets.

The performance improvement through transfer learning was not the case for the entire dataset. In general, transfer learning may improve the performance depending on the specific dataset and training configuration. Therefore, it is beneficial to test transfer learning and data augmentation techniques to improve a network’s generalization capabilities.

Marginal sports in particular can benefit from a transfer learning approach, since acquiring study participants is cumbersome.

This might also make activity recognition in marginal sports more interesting for smartwatches or fitness trackers, since the amount of data to build a solid activity recognition system is relatively small. Using activity recognition in marginal sports also increases professionalism and may thus lead to better performances in marginal sports.

## 6. Conclusions

The goal of this work was to create the first HAR system, which can classify different throwing techniques of Ultimate Frisbee players. Therefore we recorded 14 participants who performed seven different throws using a wrist-worn accelerometer and gyroscope. In order to establish a ground truth, the signal was manually annotated using a video reference. Following the action detection, a CNN was used to classify the throws and catches, and assign the false positives of the action detection to a null class. A leave-one-subject-out cross-validation resulted in a mean accuracy of 66.6%. However, the macro averaged F1 score of 52.3% is more meaningful for the imbalanced dataset. We found that separating flat, outside-in and inside-out throws is a tough challenge, and the network could not solve this task reliably. If we drop the distinction between these detailed classes and focus on the main throwing techniques and catches, the CNN reaches a macro averaged F1 score of 88.4%.

Furthermore, we studied the effects of transfer learning, using the beach volleyball dataset by Kautz et al. [8] as a source domain. We used the weights of the lower convolutional layers of a network pre-trained on the volleyball data. With all of the three convolutional layers pre-trained but not frozen, the CNN reached a comparable performance concerning the accuracy and F1 score, but achieved faster training times. Freezing more layers resulted in even shorter training times but had a slight negative impact on the prediction capabilities of the network.

Another question was whether transfer learning could improve classification tasks if only very few data samples were available. Therefore, we dropped the data augmentation steps to simulate a smaller dataset. While the smaller training dataset led to a general decrease in accuracy, the transfer learning approach improved both accuracy and F1 score. The training time slightly increases or decreases with transfer learning depending on the training configuration.

To conclude, transfer learning can be a powerful tool to combat long training times for HAR. Despite this, it should be considered in addition to data augmentation techniques to improve the generalization capabilities, especially when dealing with small datasets, which is often the case in marginal sports.

## Figures and Tables

**Figure 1 sensors-22-02560-f001:**
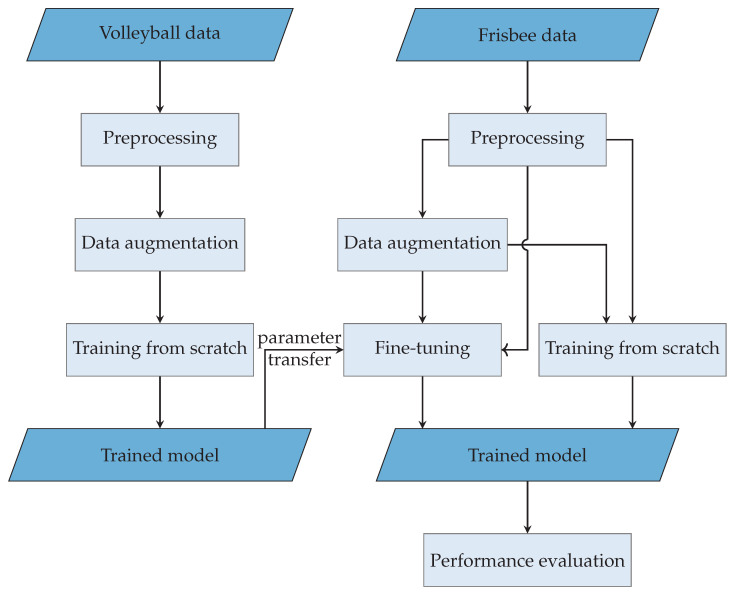
A visual summary of the workflow of our paper. We developed the first human activity recognition system for Ultimate Frisbee. Therefore, we use a convolutional neural network. Additionally, we investigated the possible improvement of the classification using a network pre-trained on volleyball activities.

**Figure 2 sensors-22-02560-f002:**
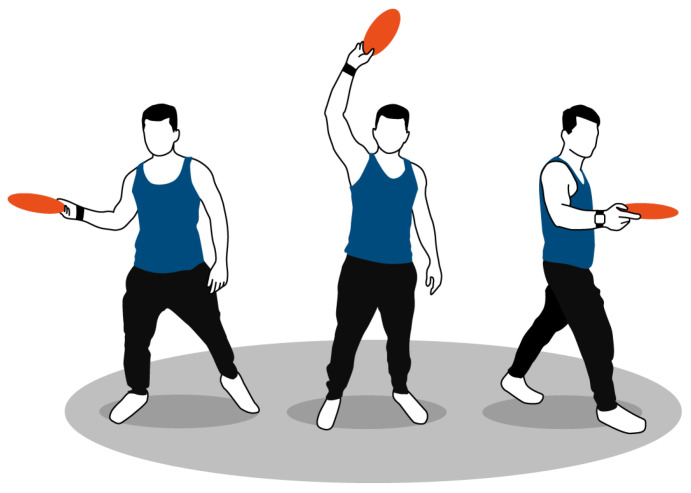
The three main throwing techniques in Ultimate Frisbee are, from left to right, forehand, overhead and backhand. The sensor is placed at the wrist of the dominant hand. The sensor (white) is fixed with a wristband (black).

**Figure 3 sensors-22-02560-f003:**
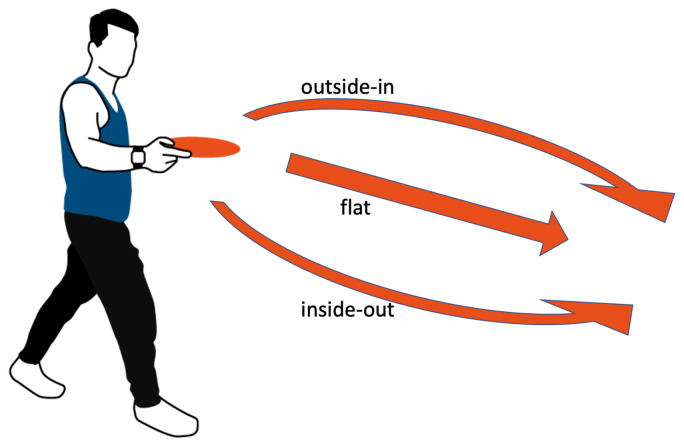
Depending on the angle of the release, the frisbee moves on a different trajectory. The different throwing types are shown for backhand throws.

**Figure 4 sensors-22-02560-f004:**
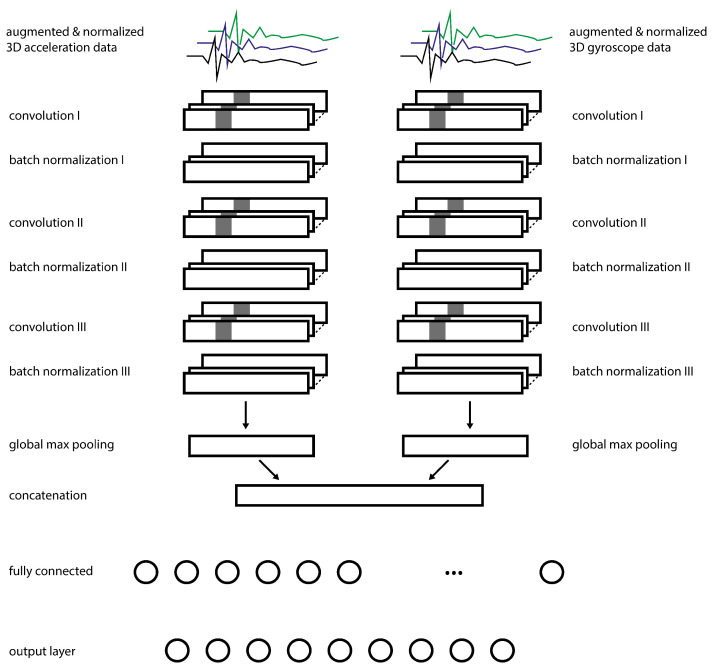
The architecture of the convolutional neural network, which was used in this work. The output dimensions of the layers are given at the top right of each row. The connections of the last two fully connected layers were omitted for better readability. The architecture was partly adapted from Fawaz et al. [29].

**Figure 5 sensors-22-02560-f005:**
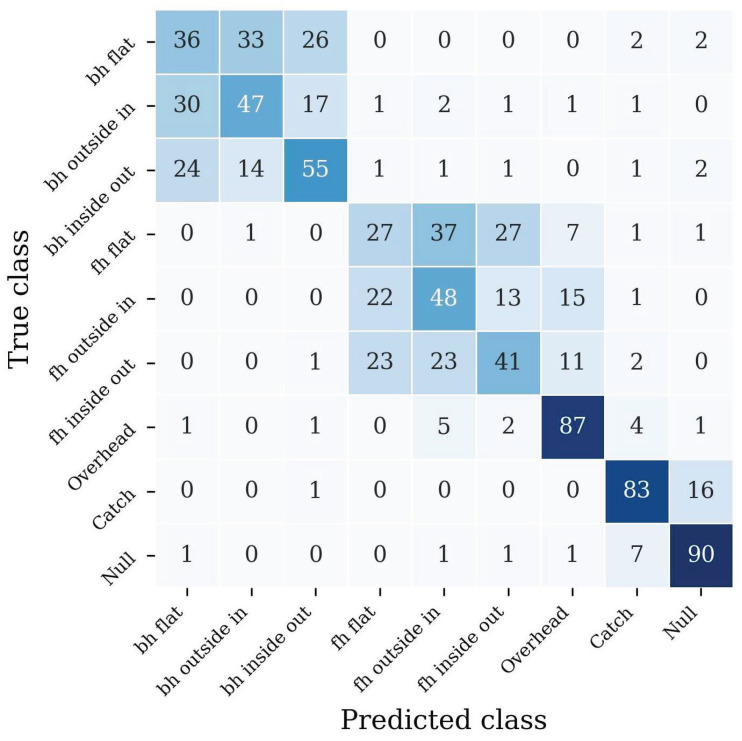
Confusion matrix of classification performance. The backhand throws are abbreviated as bh and the forehand throws as fh. The true class is given on the left and the predicted class on the bottom. Each entry is normalized according to its true class and denoted in percentage. Thus, the highest possible value is 100, and the smallest value is zero.

**Figure 6 sensors-22-02560-f006:**
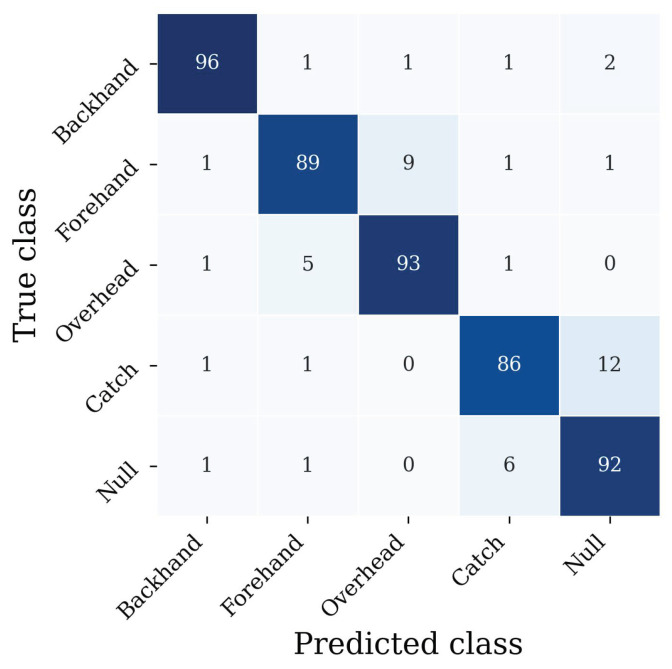
Confusion matrix for the aggregated class problem (no distinction between flat, outside-in and inside-out throws). Each entry is normalized according to the true class and denoted in percentage. Thus, the highest possible value is 100, and the smallest possible value is zero.

**Figure 7 sensors-22-02560-f007:**
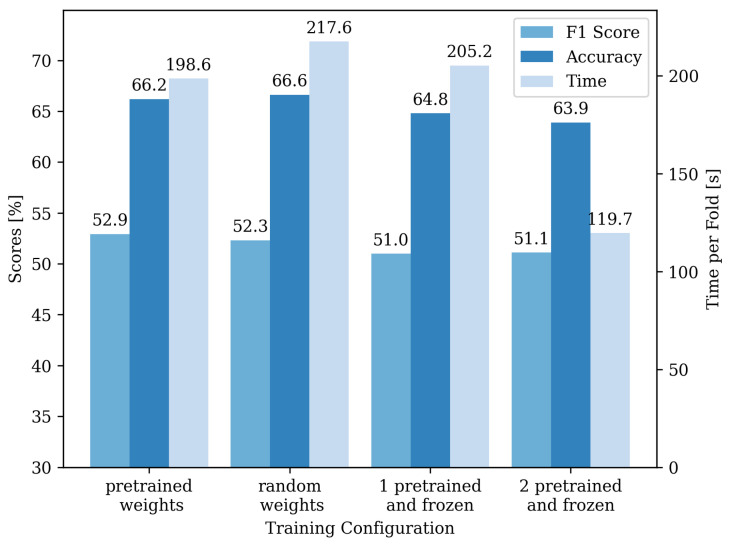
Comparison of the accuracy, F1 score, and training time for different transfer learning configurations. The scale for the accuracy and F1 score is shown on the left y-axis, and for the training time on the right y-axis. In the configuration of the pre-trained weight, all three convolutional layers were initialized with the weights of the network trained on volleyball data but not frozen. For the random weights configuration, all layers were randomly initialized. For the configurations with one or two frozen layers, the model is initialized with pre-trained weights, and the weights are frozen in the later training procedure.

**Figure 8 sensors-22-02560-f008:**
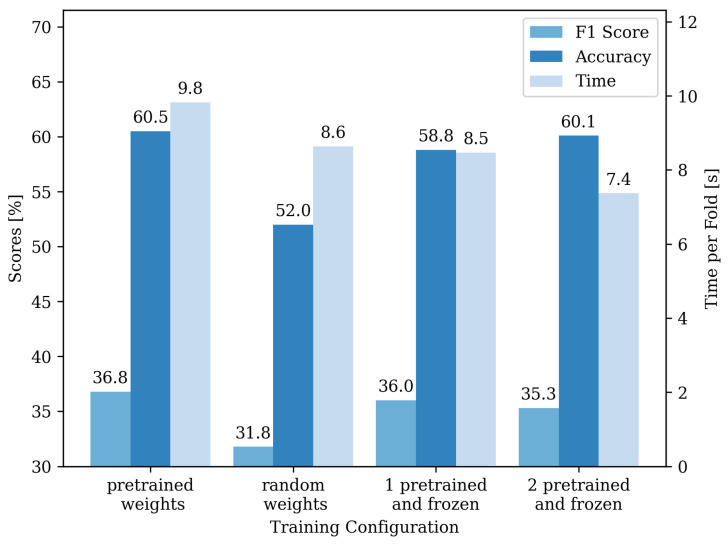
Comparison of the accuracy, F1 score, and training time for different transfer learning configurations for a reduced dataset, i.e., without data augmentation. The scale for the accuracy and F1 score is shown on the left *y*-axis and for the training time on the right *y*-axis. In the configuration of the pre-trained weights, all three convolutional layers were initialized, with the weights of the network trained on volleyball data but not frozen. For the random weights configuration, all layers were randomly initialized. For the configurations with one or two frozen layers, the model is initialized with pre-trained weights, and the weights are frozen in the later training procedure.

**Table 1 sensors-22-02560-t001:** Summary of the volleyball and frisbee dataset used in this study.

Dataset	Volleyball	Frisbee
Actions of interest (without null class)	4284	3695
Subjects (female/male)	30 (11/19)	14 (2/12)
Skill Level	Beginner to professional	Experienced amateurs
Sensor	BMA280	Portabiles NilsPod
Sampling rate	39 Hz	512 Hz downsampled to 39 Hz
Sensor placement	Wrist of dominant hand	
Reference	Manually labeled video	

**Table 2 sensors-22-02560-t002:** Description of the Ultimate Frisbee action classes used in this study for throwing with the right hand. The different angles of the frisbee during the release affect the trajectory of the frisbee, which are the namesakes for the throwing technique. Additionally, the number of samples of the respective class is specified.

Type of Action	Description	Number of Samples
Backhand flat	The athlete is standing with the shoulder axis pointing to the throwing target. The frisbee is moved from the back shoulder towards the target in front of the body. The frisbee is flat during the release. It is the most common frisbee throw.	346
Backhand outside-in	The basic movement is the same as the backhand flat but with the top of the disc facing towards the thrower during the release.	295
Backhand inside-out	The basic movement is the same as the backhand flat but with the top of the disc facing away from the thrower during the release.	305
Forehand flat	The athlete is standing with the shoulder axis perpendicular to the target. The frisbee is moved on the side of the throwing hand next to the torso. The frisbee is flat during the release.	324
Forehand outside-in	The basic movement is the same as the forehand flat but with the top of the disc facing towards the thrower during the release.	313
Forehand inside-out	The basic movement is the same as the forehand flat but with the top of the disc facing away from the thrower during the release.	307
Overhead	The athlete stands with the shoulder axis perpendicular to the target. The frisbee is moved over the head, similar to a slap shot.	341
Catch	The athlete catches the frisbee with the dominant hand or both hands.	1556
Attempted catch	The athlete tries to catch the frisbee with the dominant hand or both hands, but the frisbee bounces off or falls to the ground.	148
Null class	Non-frisbee actions during the data acquisition like running, clapping, and undefined motions.	631

**Table 3 sensors-22-02560-t003:** Detailed description of the convolutional neural network used in this study. The first convolutional together with the batch normalization layers are used once for the acceleration data and once for the gyroscope data.

Layer Type	Hyperparameter	Output Shape	# of Parameters
1D Convolution	Filter: 128, kernelsize: 8	(64, 128)	3200
Batch Normalization	Momentum: 0.9, epsilon: 0.001	(64, 128)	256
1D Convolution	Filter: 256, kernelsize: 5	(60, 128)	164,096
Batch Normalization	Momentum: 0.9, epsilon: 0.001	(60, 128)	240
1D Convolution	Filter: 128, kernelsize: 3	(58, 128)	98,432
Batch Normalization	Momentum: 0.9, epsilon: 0.001	(58, 128)	232
Global max-pooling		(128)	0
Fully Connected		(64)	16,448
Dropout	dropout: 0.2	(64)	0
Dense		(5)	325

## Data Availability

The data is available from the authors on reasonable request.

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
