# Peer review of "Wearable Sensors for Activity Recognition in Ultimate Frisbee Using Convolutional Neural Networks and Transfer Learning"

_sensors, 2022, doi:10.3390/s22072560_

Round 1

Reviewer 1 Report

In this submission, the authors recorded and annotated inertial measurement unit (IMU) data containing different types of Ultimate Frisbee throws. The relevant actions were detected by comparing the signal with the rolling mean and standard deviation The authors use a Convolutional Neural Network (CNN) and find that their proposed pipeline reaches an accuracy of 66.6% distinguishing between nine different fine-grained classes. For the classification of the three basic throwing techniques, the authors achieve an accuracy of 89.9%. The authors find that transfer learning can be beneficial, especially when dealing with small datasets, as in marginal sports.

I consider this manuscript to be of interest to researchers using machine learning techniques in transfer learning as well as readers of this journal. As such, I am somewhat in favor of publication with a few minor additions/revisions that should be included prior to publication. In particular, there has been much work on using machine learning techniques using neural network models and other data on time-varying data, which should be mentioned:

Sci. Rep. 11, 1839 (2021)
Phys. Chem. Chem. Phys., 22, 22889-22899 (2020)

In particular, these prior works also utilized complementary advanced neural network approaches to process complex data for subsequent analysis. Some of these studies also found that utilizing a variety of data-post-processing techniques could also improve accuracy. I am not asking the authors to carry out such a study, but it should be mentioned that such machine learning approaches have been used for these processes. With this minor revision, I would be willing to re-review this manuscript for possible subsequent publication.

Reviewer 2 Report

The author proposed CNN transfer learning for activity recognition using wearable sensors. Following are my concerns:

1) The abstract should clearly present the problem definition and the outcome of the research. What will be the benefit of this automation?

2) Add topic related Keywords

3) Figure 1 just represents the transfer learning, rather it should explain the overall workflow of the proposed method/ architecture. 

4) There must be some experiments with famous networks like VGG/ResNet (trained from the scratch), and there should be experiments with VGG/ResNet transfer learning  (using ImageNet weights).

5) How the authors dealt with class-imbalance 

6) Comparative analysis is required.

Round 2

Reviewer 2 Report

The authors responded to the comments adequately. I vote for acceptance of this paper in its current form.